# Supporting Infants’ Motor Development through Water Activities: A Preliminary Case–Control Study

**DOI:** 10.3390/healthcare12161556

**Published:** 2024-08-06

**Authors:** Oliwia Jakobowicz, Anna Ogonowska-Slodownik

**Affiliations:** Faculty of Rehabilitation, Jozef Pilsudski University of Physical Education in Warsaw, Marymoncka 34, 00-968 Warsaw, Poland; anna.ogonowskaslodownik@awf.edu.pl

**Keywords:** aquatic therapy, stimulation, motor development, swimming, AIMS, EMQ

## Abstract

The first twelve months of an infant’s life are the most dynamic time in the development of motor activities. Water activities can positively stimulate the motor skills, visual perception, and cognitive abilities of infants. The purpose of this study was to evaluate the motor development of infants aged 3–12 months participating in water activities. Participants in the study included 43 infants aged 3–12 months who attended water activities classes at Warsaw City swimming pools (n = 21) and infants not attending any classes (n = 22). Two methods of assessing motor development were used in the study: the Alberta Infant Motor Scale (AIMS) and the Early Motor Development Questionnaire (EMQ). The raw scores of the AIMS test and the EMQ questionnaire were statistically significantly different (*p* < 0.05) in both groups between the first and second examinations. In contrast, there was a statistically significant improvement in motor development measured by AIMS and expressed in percentiles (*p* = 0.002) and in percentiles for Polish children (*p* = 0.030) in the water group. The age-independent total score of the EMQ before and after the intervention did not change significantly (*p* = 0.149). The water environment has the potential to support the motor development of infants aged 3–12 months.

## 1. Introduction

According to the World Health Organization, approximately 52.9 million children under the age of five experience delays in the developmental process [1]. An important aspect of psychomotor development is the development of gross motor skills [2]. Each month, new motor skills are acquired that provide the means to explore the world. Acquiring a new motor skill, such as sitting, enables the infant to focus attention, manipulate objects freely, and share in a completely different perspective [3]. If an infant does not present a given motor skill until a certain age, this may indicate delayed motor development. Monitoring early motor development allows for the earlier detection of developmental delays or disorders, which prevents the appearance of possible structural and functional limitations in the future [4]. Children acquire new motor skills through learning and sensorimotor experiences such as touch, sight, hearing, smell, and taste [5]. Supporting early motor development should be holistic in nature, not only stimulating the motor system but also the child’s senses. Infant development depends on many factors, and one of them is regular and properly focused stimulation [2].

The water environment comprehensively stimulates the child’s body. Water offers a variety of conditions, but the specific interaction is the physiological and hydrodynamic effects of immersion [6]. Formal aquatic activities have positive effects on the visual motion and cognitive flexibility of infants [7]. The water’s rich stimulation creates a relaxing effect, reducing stress and anxiety [8]. In addition, water activities reduce sleep problems in children [9]. Regular participation in water activities improves neuromuscular coordination, body perception, and movement in space [10]. An indisputable advantage of water activities is the overall improvement in fitness by adapting the cardiovascular system to exercise [11]. Karpov et al. [12] highlight the health-promoting importance of swimming, which includes strengthening the immune system and preventing disease. Playing in the water positively affects the development of the senses and strengthens the child’s confidence and willingness to try new movements [13]. Infants perform exercises with their parents during classes, which affects the parent–child bond, facilitating the development of the child’s emotional sphere [14]. Classes in a group format are important and provide an opportunity for contact with peers. They allow the exchange of experiences between children and build a sense of community, increasing motivation to exercise [15]. The buoyancy of water provides support and relieves pressure on joints, allowing people with musculoskeletal dysfunctions to actively participate in water activities [16]. The use of toys and buoyancy equipment during therapy supports the acquisition of new motor skills for children with special needs [17].

Research has shown how activity in water can positively influence the acquisition of new skills in children [7,18]. Nevertheless, published studies included different forms of stimulation through water, like sports swimming, aquatic therapy, baby’s spa, and infant swimming. The difficulty of assessing the motor development of infants and the ability to work with this age group is reflected in the frequent choice to include older children in the studies. Early school-aged children who attend swimming classes are more efficient in cognitive and motor development than children who do not have swimming experience [19,20]. Early experience in water reduces the risk of developmental disorders in preschool age [21]. More scientific evidence is needed on aquatic activities for infants. In the literature review about the effects of exposure to formal aquatic activities on babies, there were 18 publications, including 6 that assessed motor development and only 3 that focused on children below 1 year old.

In our study, we wanted to assess the motor skills of infants in the first year of life. This age range was chosen because the available literature shows that younger infants have physiological characteristics that prepare them specifically for aquatic activities, such as the diving reflex [22]. In addition, a study by Eliks [23] showed that the rate of motor development varies depending on age. Difficulty in choosing appropriate research methods to assess motor development in infants might result in contradictory or unexplained conclusions [24]. In our study, we wanted to look at infants holistically, so we used a physical examination by a physiotherapist and included parents of children who completed a questionnaire on early motor development. To our knowledge, there are no studies that combine those two methods to assess the impact of water activities on infants.

The purpose of this study was to evaluate the motor development of infants aged 3–12 months participating in water activities.

## 2. Materials and Methods

This study is a prospective and controlled study involving pre- and post-intervention evaluation. The study was registered at Clinicaltrials.gov (NCT06180330; 12 December 2023). The study included non-randomized allocation and parallel-assignment methods. The study did not use masking. It was conducted from January to March 2024. It complied with the latest version of the Declaration of Helsinki and received approval from the Ethics Committee of AWF Warsaw (SKE 01-47/2023). All parents of infants were informed in detail about the objectives, conduct, and methods of the study, as well as the possible benefits and risks associated with their participation, and signed a consent form before the study was conducted.

### 2.1. Study Population

Participants in the study included 43 infants aged 3–12 months. Of the infants, n = 21 were recruited from a swimming school that organizes water classes for infants (water group), and n = 22 from places that provide classes for parents in Warsaw, in which babies did not participate in any classes (control group). The general characteristics of the infants are shown in Table 1. Because the rate of motor development varies depending on age, we also analyzed children divided into age categories.

The inclusion criteria was age of 3–12 months. Inclusion to the water group was based on regular participation in water activities once a week for 30 min. In the control group were children who did not participate in any activities supporting motor development. The study did not include children who had previously participated in the same or similar classes in water or children with motor or intellectual disabilities affecting development.

### 2.2. Assessments

Two standardized methods of assessing motor development were used in the study. First method was the Alberta Infant Motor Scale (AIMS) created by Piper and Darrah (1994). This scale is based on observation of infants’ motor development using a standardized form. The scale consists of 58 items classified into 4 main domains: lying supine (9 items), lying prone (21 items), sitting (12 items), and standing (16 items). During observation, spontaneous movements are assessed with minimal assistance. The examiner identifies the least and most mature motor skill, the so-called “developmental window”, and then rates each position of the developmental window as observed = 1 point or unobserved = 0 points. The total of all points is summed up to generate AIMS raw score, which is then placed on a centile grid [25]. The AIMS test scores were also referenced to centile grids for Polish children [23]. Because the norms for Polish children were developed recently and authors pointed out some limitations, we used original and Polish centile grids to analyze the scores.

The second method used was the Early Motor Questionnaire (EMQ) [26]. It was conducted using a pre-made form consisting of questions divided into 3 categories: gross motor, fine motor, and integration, perception, and action. This questionnaire aims to determine the motor skills of children aged between 2 and 24 months. It was filled out by parents of children in paper form in Polish [27]. Each behavior was scored from −2 to 2, where −2 means it is completely certain that the child did not exhibit the skill, −1 means that the child probably cannot perform the skill, 0 means that it is unclear whether the child has the skill, 1 means that the child probably did exhibit the skill, and 2 means that the child certainly has the skill. The scores were first summed to a raw EMQ score and then substituted into a formula for age-independent scores. Global (GL age-independent score) within T-score range of 35–65 indicates a normal level of motor development [28].

### 2.3. Intervention Protocol

The assessment was conducted before and immediately after the 9-week intervention, following the same procedure, and was performed by the same researcher (physiotherapist working with children). The clinical examination took place in a quiet, pre-prepared room equipped with a smooth mat, a recliner, and toys. Each assessment according to the AIMS took 20 min and was conducted in the presence of a parent or legal guardian. Parents were given an EMQ form to fill out, in which they answered questions referring to the infant’s age at the time of the assessment. The questionnaire was based on parents’ current knowledge of their children’s motor development.

The water activities for infants included a series of classes in water lasting 9 weeks. The classes were held once a week for 30 min each. Activities were performed by the parents under the guidance of a qualified instructor who was present in the water with the group. The water temperature ranged from 31 to 33 °C. The swimming pool measured 17 × 8 m with a maximum depth of 1.15 m. Exercises were age-appropriate, with the level of difficulty gradually increasing. The activities used during the classes are listed in Table 2.

### 2.4. Statistical Analysis

Statistical analyses were completed using Statistica 14 (TIBCO Software Inc., Palo Alto, CA, USA 2020, Data Science Workbench, version 14). A *t*-test for dependent samples was conducted to assess the significance of differences in motor development measured by AIMS before and after the intervention in each group. The Wilcoxon paired rank order test for non-parametric dependent data was used to assess the significance of differences in motor development measures using EMQ, AIMS percentile, and AIMS percentile for Polish children, as well as the sum of the age-independent score (GL) of the EMQ questionnaire, before and after the intervention in each group. Pearson’s simple correlation coefficient (r) between age and the difference in AIMS scores was calculated for the entire study group. The significance level, according to the methodology used, was set at *p* ≤ 0.05.

## 3. Results

Both groups differed significantly between the first and second assessments in terms of raw AIMS and EMQ scores (*p* = 0.001; Table 3). The mean and SD of the change in AIMS raw test scores in the water group was 12.42 ± 5.70, and those in the control group were 9.13 ± 3.89.

There was a statistically significant improvement in the AIMS percentile value (*p* = 0.002) and in the AIMS percentile for Polish children (*p* = 0.031) only in the water group (Table 4). The total score of the EMQ questionnaire independent of age was not significantly different in the aquatic and control groups after a period of 9 weeks (*p* = 0.149, *p* = 0.465, respectively). Based on EMQ age-independent results, all the infants before and after intervention were within the normal values of motor development (T-score).

Figure 1 shows the AIMS percentile score in the water group. In both infants aged 3–6 and 7–12 months, a statistically significant improvement was observed after the 9-week intervention (*p* = 0.027; *p* = 0.023, respectively).

Figure 2 shows the Alberta Infant Motor Scale percentile score for Polish children in the water group. The analysis showed statistically significant changes only in infants aged 7–12 months (*p* = 0.042).

Table 5 shows the result of the correlation between the age of the infants and the difference in scores on the AIMS test before and after the intervention. The analysis showed a statistically significant negative correlation between the variables.

## 4. Discussion

The aquatic environment can support the acquisition and improvement of motor skills in early infancy [18]. The topic of supporting infants’ motor development through water activities is gaining more attention. The results of our study showed that infants naturally acquire motor skills as they grow older, regardless of participation in the intervention. This was supported by the raw AIMS and EMQ scores, which changed significantly between the first and second tests in both groups. However, only the infants in the water group had a statistically significant improvement in their AIMS score expressed in percentiles. Similar results were obtained by Dias et al. [24], who observed an improvement in the raw AIMS score in the study groups, with a noticeable change in the AIMS percentile ranks only in the experimental groups.

The positive effect of water on the development of gross motor skills was demonstrated in a study by Boroni et al. [29], which showed that a ten-week intervention was sufficient enough to increase the level of motor skills in children. In addition, the form of water activities had a stimulating effect on infants’ cognitive development. The benefits of water activities were also confirmed in a study conducted by Duda et al. based on parents’ reports. The parents indicated that water activities had a stimulating effect on the psychomotor development of children aged 6 months to 3 years. Moreover, water facilitated infants in acquiring new motor skills [30]. An interesting study by Araujo et al. [31] examined the effects of the Kids Intervention Therapy—Aquatic program on the neuro-psychomotor development of infants aged 4–18 months. Their study showed that activities in an aquatic environment had a positive effect on both typically developing and those with delayed motor development, including premature infants [31]. Infants receiving additional stimulation through swimming showed improved visual perception and cognitive abilities [32]. Additionally, children were calmer, more relaxed, and cried less after water activities [33].

AIMS is a standardized scale used to assess the motor skills of infants. It was developed and standardized in 1994 by Piper and Darrah of the University of Alberta, Canada. Over time, differences have been noted in the level of acquired motor skills of infants from different populations in relation to Canadian norms. A study of Polish infants showed that mean AIMS raw scores were similar to or lower than Canadian values in most age groups [23]. Significantly lower values were found in children in the first two months of life and in those between 4–7 months and 13–16 months. These studies suggested that the Canadian norms may not be fully adequate for Polish children. Our study showed that none of the children had motor skills at the 90th percentile according to Canadian norms, while according to norms for Polish infants, there were four children in the 90th percentile. Additionally, only children aged 7–12 months in the water group showed improvement in the AIMS test score expressed in percentiles for Polish children, confirming the discrepancy between centile grids in different countries.

Our study showed that younger children were characterized by a faster rate of motor development. This regularity is confirmed by a study conducted by Pereira et al. [34], in which infants were attending aquatic classes. An analysis of the study using AIMS not only demonstrated the positive effect of water on infant development but also concluded that earlier infant contact with water is associated with better motor skills in early childhood. Sigmundsson and Hopkins [21] showed that four-year-old children who participated in water activities as infants had better grasping function and static balance compared to their peers.

A child’s motor development can be assessed with tests conducted by specialists or with parent-reported measurements. Specialists are well aware of the testing protocol and typical child development, but the test outcome depends significantly on the comfort level and relationship between the child and the specialist. Parents’ reports on their children have shown good accuracy [35,36]. The EMQ questionnaire provided by parents is a reliable and inexpensive assessment of motor development [26]. In our study, the EMQ questionnaire scores were first summed to the raw EMQ score and then converted to age-independent scores. This allowed a direct comparison of infants of different ages and facilitated the identification of infants who scored above and below the expected average. Our results showed no significant differences between infants during the first and second tests and no differences between groups. Additionally, before and after the 9-week intervention, the age-independent score of the EMQ questionnaire for all infants was within normal limits (35–65) [28].

Improvements in motor skills in the raw scores in both groups measured with AIMS and EMQ occurred after 9 weeks, which showed the natural acquisition of motor skills with age. Looking at the age-independent scores in the water group, a statistically significant change occurred only in the AIMS results. The reason for this may be that, independent of age, the EMQ questionnaire score is more stable over time [28]. Libertus and Landa [26] compared an EMQ questionnaire completed by parents with a specialist’s assessment using Peabody Developmental Motor Scales (PDMS-2). A comparison of the EMQ with the PDSM-2 showed a negative relationship between parent and specialist ratings [26]. Although questionnaires completed by parents are a reliable method of assessment, it is important to remember that parents do not have specialized skills to assess motor development, so they may overestimate or underestimate their child’s skills.

### Limitations

It is important to note several limitations of this study. Despite the fact that the study attempted to be carried out in homogeneous groups (healthy children with normal motor development), there remain a few factors that may have impacted the results. Other stimulation of development performed by parents and standard of living were not assessed. In addition, the study group was small, and the evaluation of infants was performed only twice. A possible extension of the intervention, even to a six-month or one-year follow-up, would allow an assessment of the long-term effects of water activities on motor development.

## 5. Conclusions

The participation of infants aged 3–12 months in water activities supports their early motor development. There is a need to create a core outcome set as a guideline to assessing motor development in infants.

## Figures and Tables

**Figure 1 healthcare-12-01556-f001:**
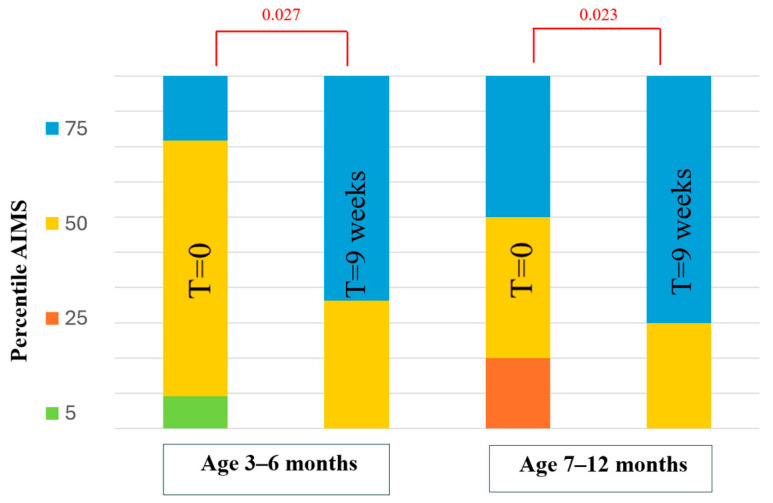
Alberta Infant Motor Scale percentile in the water group.

**Figure 2 healthcare-12-01556-f002:**
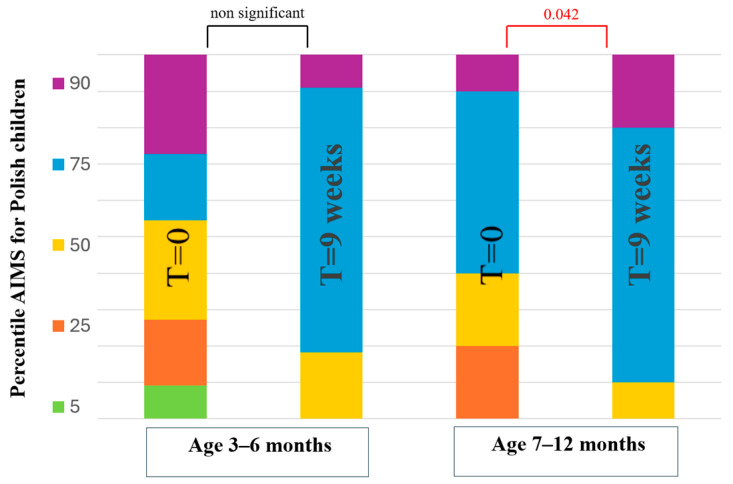
Alberta Infant Motor Scale percentile for Polish children in the water group.

**Table 1 healthcare-12-01556-t001:** Characteristics of the studied infants.

Variable	Group	Mean (SD)	*p*
Age (days)	Water	208.95 (51.81)	0.799
	Control	196.86 (75.23)	
Birth weight (g)	Water	3338.10 (509.39)	0.545
	Control	3375.00 (403.77)	
Birth body height (cm)	Water	52.86 (3.89)	0.690
	Control	53.30 (3.10)	
Girls/boys (number)	Water	9/12	
	Control	11/11	
Age 3–6 months (number)	Water	11	
	Control	12	
Age 7–12 months (number)	Water	10	
	Control	10	

**Table 2 healthcare-12-01556-t002:** Examples of exercises performed during the water activities for infants.

Exercise	Organizational Notes	Number of Repetitions/Exercise Time
Introductory part		
Song of greeting	Infant is held under the arms vertically to the surface of the water. Parents with infants face the center of the circle. Everybody sings a greeting song.	1′
Greeting children	Infant is held under the arms, horizontal to the surface of the water. Parents with infants face the center of the circle. At the word “greeting”, parents come to the center, greet their babies, and return to their places.	1′
Swimming on the belly	Infant is held under the arms horizontally to the surface of the water. Parents with infants move around the perimeter of the circle. At the instructor’s signal, they change direction.	1′
Rotations on the belly	Infant is held under the arms horizontally to the surface of the water. Parents with infants move around the perimeter of the circle performing body tilts in the frontal plane.	1′
Jumps	Infant is held under the arms. Parent and infant face each other. Parents with infants move around the perimeter of the circle. At the signal of the instructor, the parents lift the baby up. *We observed sensitivity to vestibular stimulation.	1′
Main part		
Flying balls	Infant is held under the arms horizontally to the surface of the water. The instructor counts down 1, 2, 3 and throws colored balls into the pool. The infants grab the ball.	1′
Treasure	Infant is held under the arms horizontally to the surface of the water facing the direction of the swim. The infants grab the ball with both hands and then throw it into the box.	1′
Back swimming	Infant is held under thighs and lies on their back, with head resting on the parent’s shoulder. The parent walks backwards in the direction of the swim and shows the child a book.	1 length of the pool
Rotation and toss	Infant is held under the arms vertically to the surface of the water. The parent performs a 360° turn, then lifts the child upward. *It is important that the parent performs the rotation in both directions.	1 length of the pool
Bubbles	Infant is supported under chest horizontally to the surface of the water. Parent and infant face each other. The parent moves backwards in the direction of the swim and shows the infant the water bubbles.	1 length of the pool
Motorboats on the back	Infant is supported at shoulder height. The parent walks backward in the direction of the swim and performs a frontal plane tilt with the infant.	1 length of the pool
Snaking	Infant is held under the arms. Parent and infant face each other in the direction of the swim, performing frontal plane body tilts and gliding through the water.	1 length of the pool
Catch and throw	Infant is held under the arms, horizontally to the surface of the water. The parent throws the ball forward, they move together towards it so that the baby can catch it.	1 length of the pool
Throw and jump	Infant is held under the arms, horizontally to the surface of the water. Parent throws the ball forward, counts down 1, 2, 3, and jumps on the ball.	1 length of the pool
Shot at the goal	Infant is held under thighs, lying on back, head resting on the parent’s shoulder. Parent kicks the ball with the child’s legs to make the ball move forward. At the end of the length, they must hit the inflatable goal.	1 length of the pool
Ball massage	Infant is held on forearm, horizontally to the surface of the water. Parent massages the baby’s back with a ball with spikes.	1 length of the pool
Floating mat on the belly	Infant lies forward on the mat. The child assumes proper forearm support or enters a quadruped position. Parent pours water from a cup over the child’s back and head.	1 length of the pool
Rotation	Infant lies on the mat on the back and learns the correct rotation from the back to belly.	1 length of the pool
Floating mat on the back	Infant lies on the mat on the back. Parent holds the mat with one hand, moves backward with the other hand, holding a mirror over the child’s face, showing the child’s reflection.	1 length of the pool
Jump off the mat	Infant lies forward on the mat. The child assumes proper forearm support or enters a quadruped position. Parent sings a song and rocks the mat, then finally performs a jump into the water.	1 length of the pool
Pouring	Infant sits or lies on the shore. The parent pours water over the baby’s feet back and head with a watering can.	5 repetitions
Introduction to diving	Infant is held on forearm, vertically to the surface of the water. Parent clearly says the child’s name at the phrase “we dive”. The parent pours a whole bucket of water over the infant’s head.	3 repetitions
Diver	Infant is held under the arms. Parent and child face each other. Parent sings a song that ends with the baby being submerged underwater for 2 s.	1 repetition
Cool down		
Goodbye song	Infant held under the arms, vertically to the surface of the water. Parents and their infants face the center of the circle. Everybody sing a goodbye song.	1′
Silence	Infant held under the arms, as parent and infant face each other. Parent rocks and hugs the infant.	1′

**Table 3 healthcare-12-01556-t003:** Motor development scores on the AIMS scale before and after infant intervention.

		Pre	Post	t (df)	*p*	95% CI
AIMS raw test score ^a^						
Water group	Mean	31.00	43.43	−9.98 (20)	0.001	−15.02 to −9.83
	SD	11.45	9.13			
Control group	Mean	30.00	39.14	−11.00 (21)	0.001	−10.86 to −7.40
	SD	13.71	12.04			
EMQ raw test score ^b^						
Water group	Mean	−74.19	−32.95	NA	0.001	NA
	SD	67.33	68.18			
Control group	Mean	−99.86	−53.09	NA	0.001	NA
	SD	67.74	71.30			

AIMS—Alberta Infant Motor Scale, EMQ—Early Motor Questionnaire, NA—not applicable; ^a^ analyzed with *t*-test for dependent samples; ^b^ analyzed with Wilcoxon paired rank order test.

**Table 4 healthcare-12-01556-t004:** Results of infant motor development parameters.

Variable	Water Group (n = 21)				Control Group (n = 22)			
	Pre	Post	Z	*p*	Pre	Post	Z	*p*
AIMS percentile								
Mean	51.43	66.67	3.059	0.002	65.45	65.45	0.001	1.000
SD	17.83	12.08			13.53	13.53		
Min	5	50			50	50		
Max	75	75			90	90		
AIMS percentile for Polish children								
Mean	56.67	73.1	2.158	0.031	73.18	71.59	0.625	0.532
SD	26.8	12.89			14.76	14.83		
Min	5	50			25	50		
Max	90	90			90	90		
GL age-independent score								
Mean	49.47	49.53	1.442	0.149	48.22	48.66	0.730	0.465
SD	4.89	3.69			2.67	2.4		
Min	44.87	45.16			43.56	44.21		
Max	65.44	64.07			52.60	54.59		

AIMS—Alberta Infant Motor Scale; GL—Global score.

**Table 5 healthcare-12-01556-t005:** The correlation score of age and difference of AIMS scores for all infants.

**Age and** **AIMS Difference**	**Mean (SD)**	**r(X,Y)**	**r2**	**t**	** *p* **
	202.76 (64.38)	−0.349	0.121	−2.384	0.021
	10.74 (5.08)				

AIMS—Alberta Infant Motor Scale.

## Data Availability

The original contributions presented in the study are included in the article, further inquiries can be directed to the corresponding author/s.

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
