# Peer review of "Supporting Infants’ Motor Development through Water Activities: A Preliminary Case–Control Study"

_healthcare, 2024, doi:10.3390/healthcare12161556_

Round 1

Reviewer 1 Report

Comments and Suggestions for Authors

Introduction

The aim of the study reported was to analyze the effect of baby swimming classes on the motor development of infants younger than 2 years old. Yet, in the Introduction section, the authors do not elaborate on what is already known about the influence of the attendance of aquatic activities on infants’ motor development. The studies addressing the issue are first cited in the Discussion section. Therefore, by reading the introduction of the paper, it is not clear why was the study conducted, what is known and that is the gap in the literature that the authors are trying to fill.

The authors finish the introduction by stating that “The purpose of this study was to evaluate the motor development of infants aged 3-12 months participating in water activities”. However, they never say why they have done the study. As the authors show in the Discussion section, there are already studies showing the positive effect of aquatic activities on infants’ motor development, so what is the novelty of the present study?

In line 43, where it is written "Aquatic therapy has positive effects on the motor, visual perception and cognitive abilities of infants [7]. ", it should be written, "Formal aquatic activities have positive effects on the motor, visual perception and cognitive abilities of infants [7]. ". In Santos et al. (2023) there was no study with aquatic therapies as an intervention showing improvement in visual perception or cognitive abilities reported.

In line 46, the sentence “Regular participation in water activities improves neuromuscular coordination, body perception and movement in space.” needs a reference.

In line 54, the sentence “The water's rich stimulation creates a relaxing effect, reducing stress and anxiety.” needs a reference.

In lines 55/56, the sentence “Infants perform exercises with their parents during classes, which affects the parent-child bond facilitating the development of the child's emotional sphere.” needs a reference.

Materials and Methods

The first sentence of the Results section and Table 2 should be in the 2.1 Study Population section.

2.2 Assessments

In line 88, the authors state “In addition, the AIMS test scores were referenced to centile grids for Polish children [17]”. If the population of the study was Polish, why would the authors use the Canadian scale in the first place if there are norms for a Polish AIMS version?

2.3 Intervention Protocol 

In line 102, it is written: “The assessment was performed before the intervention and after 9 weeks”. It is not clear “after” what. After 9 weeks of the end of the intervention or after 9 weeks of the first test? 

Why compare the AIMS scores of the Polish population with the Canadian population? Eliks and colleagues (2023) had already shown that AIMS total scores in the Polish population were significantly lower than in the Canadian population, so it is not a surprise that in the present study, the AIMS scores were significantly lower when compared to the Canadian.

Results

The first sentence and Table 2 should be in the Materials and Methods section, categorizing the population of the study.

Discussion

The authors should elaborate on why there was a significant difference in motor development before and after the intervention when motor development was accessed with AIMS but not when accessed using the EMQ questionnaire.

The authors conclude the Discussion section by stating: “However, because of the few studies, there is no clear answer to the question of whether infant water activities affect the rate of infant motor development.”. The authors cited previous studies showing that aquatic activities improve motor development, and the present study shows the same. Why is the positive impact of aquatic activities on infants’ motor development not clear yet? In the Conclusion section, the authors state: “Participation of infants aged 3-12 months in water activities supports their early motor development.”. I would suggest changing that last sentence of the Discussion section.

Limitations

The authors should point out as a study limitation the small sample size.

Author Response

 Reply to Reviewers

Supporting infants' motor development through water activities

(Manuscript ID healthcare-3104561)

Thank you for a detailed analysis of our work and for all the comments which will undoubtedly lead to increasing the value of this article.The changes suggested by the Reviewers have been enlisted below. In order to facilitate the assessment of the changes made in the revised version of the manuscript, they have been highlighted in yellow. We hope that the changes will be satisfactory for the Reviewers and will make the paper methodologically correct and interesting for the readers.

Comments:

Introduction

1) The aim of the study reported was to analyze the effect of baby swimming classes on the motor development of infants younger than 2 years old. Yet, in the Introduction section, the authors do not elaborate on what is already known about the influence of the attendance of aquatic activities on infants’ motor development. The studies addressing the issue are first cited in the Discussion section. Therefore, by reading the introduction of the paper, it is not clear why was the study conducted, what is known and that is the gap in the literature that the authors are trying to fill. The authors finish the introduction by stating that “The purpose of this study was to evaluate the motor development of infants aged 3-12 months participating in water activities”. However, they never say why they have done the study. As the authors show in the Discussion section, there are already studies showing the positive effect of aquatic activities on infants’ motor development, so what is the novelty of the present study?

Thank you for your comment. The impact of participation in water activities and motor development in published studies has been added in the Introduction section (lines 60-82) . The purpose of this study was to evaluate the motor development of infants aged 3-12 months participating in water activities. Based on the Santos (2023) systematic review, there are only 18 studies on the effects of the aquatic environment on children's development, 6 of which are on motor development and only 3 focused on children under 1 year. In addition none of them used different methods to assess the motor development.

The study on the possibility of influencing the rate of motor development of children can be a useful working guideline for physiotherapists, specializing in the treatment of the youngest patients but also for parents who want the best possible motor start for their children.

2) In line 43, where it is written "Aquatic therapy has positive effects on the motor, visual perception and cognitive abilities of infants [7]. ", it should be written, "Formal aquatic activities have positive effects on the motor, visual perception and cognitive abilities of infants [7]. ". In Santos et al. (2023) there was no study with aquatic therapies as an intervention showing improvement in visual perception or cognitive abilities reported.

It was corrected (lines 42-42).

3) In line 46, the sentence “Regular participation in water activities improves neuromuscular coordination, body perception and movement in space.” needs a reference.

 The reference was added (line 46).

4) In line 54, the sentence “The water's rich stimulation creates a relaxing effect, reducing stress and anxiety.” needs a reference.

The reference was added (line 44).

5) In lines 55/56, the sentence “Infants perform exercises with their parents during classes, which affects the parent-child bond facilitating the development of the child's emotional sphere.” needs a reference

The reference was added (line 53).

 Materials and Methods

6) The first sentence of the Results section and Table 2 should be in the 2.1 Study Population section.

It was corrected.

7) In line 88, the authors state “In addition, the AIMS test scores were referenced to centile grids for Polish children [17]”. If the population of the study was Polish, why would the authors use the Canadian scale in the first place if there are norms for a Polish AIMS version?/// Why compare the AIMS scores of the Polish population with the Canadian population? Eliks and colleagues (2023) had already shown that AIMS total scores in the Polish population were significantly lower than in the Canadian population, so it is not a surprise that in the present study, the AIMS scores were significantly lower when compared to the Canadian.

Thank you for your comment. Although there are centile grids for Polish infants (Eliks 2023). The Polish norms are recently developed and the authors pointed out some limitations that their study was conducted in single center, located in one of Poland's largest cities, which limited the diversity of participants (e.g., the lack of people from rural areas or other districts of the country), and that the results refer only to infants born at term, so research is needed to standardize the AIMS in Polish premature infants. For that reason we presented the results on the original AIMS centile grids and also on the Polish grids.

8) In line 102, it is written: “The assessment was performed before the intervention and after 9 weeks”. It is not clear “after” what. After 9 weeks of the end of the intervention or after 9 weeks of the first test?  

It was corrected (line 136-137).

Discussion

9) The authors should elaborate on why there was a significant difference in motor development before and after the intervention when motor development was accessed with AIMS but not when accessed using the EMQ questionnaire.

It was further elaborated in the discussion (line 259-269).

10) The authors conclude the Discussion section by stating: “However, because of the few studies, there is no clear answer to the question of whether infant water activities affect the rate of infant motor development.”. The authors cited previous studies showing that aquatic activities improve motor development, and the present study shows the same. Why is the positive impact of aquatic activities on infants’ motor development not clear yet? In the Conclusion section, the authors state: “Participation of infants aged 3-12 months in water activities supports their early motor development.”. I would suggest changing that last sentence of the Discussion section.

It was corrected. Previous research, although confirming the positive effects of water on child development, includes various forms of stimulation by water, including sports swimming, water therapy, baby spa and baby swimming. We tried to make our study focused on motor skill learning on purpose and included on a 9-week plan prepared by a physiotherapist. In our project, we wanted to look at infants holistically, so we used a physical examination by a physiotherapist and included parents of children who completed a questionnaire on early motor development- this information was added in the introduction.

Limitations

11) The authors should point out as a study limitation the small sample size.

You are right it was added (line 275).

Reviewer 2 Report

Comments and Suggestions for Authors

Although I believe that the variety of conditions we offer children during development is very important, I do not believe that the work presented here can in any way objectively account for the impact of water activities. I see several major limitations in the work:

- the sample of children is very small - especially when knowing the inter-individual variability in the dynamics of children's psychomotor development

- 30 minutes of water activities per week is a low intensity compared to all the influences in the remaining 167.5 hours of the week, which were not surveyed in any way

- it is not at all clear from the methodology why the results split children into ages 3-6 months and 7-12 months - is this the age at the start of the intervention? Then the actual range of ages of children for the start of the intervention could be up to 10 months! For children in such a range, completely different water activities are recommended - with which is related a second very serious reminder. It is not possible for children in such a large age range to have the same programme - the programme in Table 1 mentions activities that are not available and even risky for a child under 4-5 months of age at this age (inability to straighten the cervical spine and hold the head upright, kicking or catching).

- The mean of all children's scores is misleading, as a more interesting statistic might be the mean of individual changes between the initial and final tests.

Although AIMS are used to compare the apparent effect, the very brief conclusion mentions the need for national standards, but the methodology refers to existing standards for Polish children (line 200).

Author Response

Reply to Reviewers

Supporting infants' motor development through water activities

(Manuscript ID healthcare-3104561)

 Thank you for a detailed analysis of our work and for all the comments which will undoubtedly lead to increasing the value of this article. The changes suggested by the Reviewers have been enlisted below. In order to facilitate the assessment of the changes made in the revised version of the manuscript, they have been highlighted in yellow. We hope that the changes will be satisfactory for the Reviewers and will make the paper methodologically correct and interesting for the readers.

Comments:

1 and 2) the sample of children is very small - especially when knowing the inter-individual variability in the dynamics of children's psychomotor development

- 30 minutes of water activities per week is a low intensity compared to all the influences in the remaining 167.5 hours of the week, which were not surveyed in any way

Thank you for your comment. We added the small sample in our limitations of the study. We believe that the study on the possibility of influencing the rate of motor development of children can be a useful working guidelines for physiotherapists, specializing in the treatment of the youngest patients but also for parents who want the best possible motor start for their children. Our study showed that even this small stimulus (30 min once a week) was able to support the motor development of the child.

3) it is not at all clear from the methodology why the results split children into ages 3-6 months and 7-12 months - is this the age at the start of the intervention? Then the actual range of ages of children for the start of the intervention could be up to 10 months!  

Thank you for your comment. This information was added (line 99-100).We decided on this division because the available literature shows that younger infants have physiological characteristics that prepare them specifically for this type of activity. Study by García (2015) showed 100% of infants aged 2 to 6 months had a diving reflex In addition, a study by Eliks (2023) showed that infants have different rate of motor development depending on the age.

4) It is not possible for children in such a large age range to have the same programme - the programme in Table 1 mentions activities that are not available and even risky for a child under 4-5 months of age at this age (inability to straighten the cervical spine and hold the head upright, kicking or catching).

Thank you for your comment. Table 1 shows examples of exercise plan used during classes for infants in water. Not all exercises were used in every class. In addition classes in water were designed to stimulate infants to acquire new motor skills, so stimulation of for example, grasping with round objects is advisable even in a child who does not have this skill. Proper grasping close to the hip joint allows safe stimulation of kicking lower limbs. A prerequisite for active participation in the water activity is the need to keep the head in a stable upright position. Newborns and small infants should be held as horizontally as possible, and with increasing head control can be positioned more vertically, so for smaller children a cervical spine stabilizing grip is used.

4) The mean of all children's scores is misleading, as a more interesting statistic might be the mean of individual changes between the initial and final tests.

Thank you for your comment. It was added according to the suggestion (lines 166-167).

5) Although AIMS are used to compare the apparent effect, the very brief conclusion mentions the need for national standards, but the methodology refers to existing standards for Polish children (line 200).

Although there are centile grids for Polish infants (Eliks 2023). The Polish norms are recently developed and the authors pointed out some limitations that their study was conducted in single center, located in one of Poland's largest cities, which limited the diversity of participants (e.g., the lack of people from rural areas or other districts of the country), and that the results refer only to infants born at term, so research is needed to standardize the AIMS in Polish premature infants. For that reason we presented the results on the original AIMS centile grids and also on the Polish grids.

Reviewer 3 Report

Comments and Suggestions for Authors

Healthcare

Manuscript: healthcare-3104561

Supporting infant’s motor development through water activities

Thank you for the opportunity to review this manuscript. It is an interesting study, and my few comments are stated below.

Abstract:

Concise and clear.

Introduction:

The authors present a pertinent review of the litterature, and state clearly the objective of their study.

Methods:

Intervention study, registered clinical trial. The authors used a motor assessment scale – AIMS – and a questionnaire EMQ completed by the parents, applied before and 9 weeks after the intervention. They described the details of the intervention.

1. But I missed in the selection criteria, the definitions of inclusion criteria, such as the age of the children to be enrolled and distribution of gender; the exclusion criteria is clear.

2. I also missed the allocation description of the children to be enrolled in each one of the two groups, and the blindness criteria used.

These informations were given in the result section, but they should be stated in Methods.

Results:

Well presented.

Discussion:

Well written

Author Response

Reply to Reviewers

Supporting infants' motor development through water activities

(Manuscript ID healthcare-3104561)

Thank you for a detailed analysis of our work and for all the comments which will undoubtedly lead to increasing the value of this article.The changes suggested by the Reviewers have been enlisted below. In order to facilitate the assessment of the changes made in the revised version of the manuscript, they have been highlighted in yellow. We hope that the changes will be satisfactory for the Reviewers and will make the paper methodologically correct and interesting for the readers.

Comments:

1) I missed in the selection criteria:

-the definitions of inclusion criteria, such as the age of the children to be enrolled and distribution of gender

It was added (lines 101-104).

-I also missed the allocation description of the children to be enrolled in each one of the two groups, and the blindness criteria used.

The children were recruited based on their participation in water activities and we did not allocate them. That is why there was no blinding in the intervention allocation.

-These informations were given in the result section, but they should be stated in Methods.

It was corrected.